# Exercise and Nutrition Strategies for Combating Sarcopenia and Type 2 Diabetes Mellitus in Older Adults

**DOI:** 10.3390/jfmk7020048

**Published:** 2022-06-08

**Authors:** Dionysia Argyropoulou, Nikolaos D. Geladas, Tzortzis Nomikos, Vassilis Paschalis

**Affiliations:** 1School of Physical Education and Sport Science, National and Kapodistrian University of Athens, 17237 Athens, Greece; argdiona@phed.uoa.gr (D.A.); ngeladas@phed.uoa.gr (N.D.G.); 2Department of Nutrition and Dietetics, Harokopio University, 17676 Athens, Greece; tnomikos@hua.gr

**Keywords:** diabetes mellitus, diet, exercise, sarcopenia

## Abstract

Medical and technology development have drastically the improved quality of life and, consequently, life expectancy. Nevertheless, the more people who enter the third-age, the more geriatric syndromes expand in the elderly. Sarcopenia and Type 2 diabetes mellitus (T2DM) are common diseases among the elderly and the literature has extensively studied these two diseases separately. Recent evidence, however, revealed that there is a bidirectional relationship between sarcopenia and T2DM. The aims of the present review were: (1) to present diet and exercise interventions for the management of sarcopenia and T2DM and (2) identify which diet and exercise interventions can be used simultaneously in order to effectively deal with these two disorders. Exercise and a balanced diet are used as effective countermeasures for combating sarcopenia and T2DM in older adults based on their bidirectional relationship. Lifestyle changes such as exercise and a balanced diet seem to play an important role in the remission of the diseases. Results showed that chronic exercise can help towards glycemic regulation as well as decrease the incidence rate of muscle degradation, while diet interventions which focus on protein or amino acids seem to successfully treat both disorders. Despite the fact that there are limited studies that deal with both disorders, it seems that a combined exercise regime (aerobic and resistance) along with protein intake > 1gr/kg/d is the safest strategy to follow in order to manage sarcopenia and T2DM concurrently.

## 1. Introduction

Skeletal muscle loss is an expected outcome of the aging process and is considered to be a significant factor in strength deterioration, loss of functional mobility, and independence [1,2]. While the causes for muscle loss are many, sarcopenia describes the progressive and generalized skeletal muscle disorder involving the accelerated loss of muscle mass and function. Sarcopenia is a term derived from the Greek phrase “poverty (penia) of flesh (sarco)” and is defined as the progressive loss of skeletal muscle and strength due to the age-related disability to keep muscular metabolism in equilibrium, affecting quality of life, fragility, and mortality. The term sarcopenia is used to differentiate this condition from other causes of muscle loss such as immobility or neurological damage [3,4]. Sarcopenia is gradually identified as a common disease in older adults, and it is among the major public health problems that must be addressed [5].

Despite the precise definition for sarcopenia, there have been multiple criteria for identifying this condition. Consensus statements regarding diagnostic criteria have been developed by European Working Group on Sarcopenia in Older People (EWGSOP) [6], the International Working Group on Sarcopenia (IWGS) [7], the Asian Working Group for Sarcopenia (AWGS) [8], and the American Foundation for the National Institutes of Health (FNIH) [9]. According to the EWGSOP, sarcopenia can be classified in prosarcopenia, sarcopenia, and severe sarcopenia based on muscle mass, strength, and physical performance. Older adults suffering from prosarcopenia exhibit only a lower muscle mass volume when compared to average healthy older adults, while individuals suffering from sarcopenia, apart from low muscle mass, also exhibit reduced strength or physical performance. Finally, in severe sarcopenia, all three factors are present. Sarcopenia can also be classified according to its duration (i.e., acute or chronic). Acute sarcopenia is caused by disease or injury and lasts no more than 6 months. Chronic sarcopenia is caused by progressive diseases and is suggested to last more than 6 months. The diagnosis of sarcopenia can be easily confused with other diseases that can lead to muscle loss (i.e., cachexia, frailty, sarcopenic obesity) [3,10]. Since the prevalence of sarcopenia largely depends on diagnostic clinical criteria, it varies vastly among different studies [11,12,13,14,15,16,17]. It should be noted that reduced muscle strength is not necessarily accompanied by reduced skeletal muscle mass and, therefore, EWGSOP precedes muscle functioning assessment to muscle mass loss as tools for diagnosing sarcopenia [12]. Specifically, the criteria for diagnosing sarcopenia should include the examination of muscle functioning and muscle mobility. Of interest, the physical status of the individual in entering the third-age is a critical factor for developing sarcopenia [18,19].

Diabetes mellitus is a common disease affecting older adults, coined as a metabolic disorder characterized by ineffective insulin secretion, defective insulin action, or both. Diabetes mellitus can be caused either by the insufficient secretion of insulin (Type 1) or/and by development of insulin resistance (Type 2). In skeletal muscles, insulin binds to the insulin receptor (GLUT4) and allows glucose to be absorbed by the muscle cells. In case of insulin resistance, there is a decrease in glucose absorption by the muscle tissue, which in turn leads to increased blood glucose levels while there is a shortage of energy in the skeletal muscle [20,21]. In 2017, approximately 462 million individuals were affected by Type 2 diabetes mellitus (T2DM) corresponding to 6.28% of the world’s population (4.4% of those aged 15–49 years, 15% of those aged 50–69, and 22% of those aged 70+) [22].

Sarcopenia is a common trait for older adults suffering from diabetes mellitus. Recent studies suggested that sarcopenia may not only be a cause for diabetes but also a consequence, indicating a bidirectional relationship between the two conditions [23,24]. Specifically, it has been found that both men and women with diabetes had low values of skeletal mass index [23], while it was found that the frequency of T2DM was higher in sarcopenic subjects when compared to their non-sarcopenic counterparts [24]. Lifestyle changes can serve in the prevention and treatment of both sarcopenia and diabetes in older adults.

Exercise, either aerobic or resistance, is widely used to combat degrading consequences of the aging process in older adults [25]. Indeed, participation in a regular exercise program can help people to improve glycemic control, to preserve their cardiovascular function, to reduce insulin resistance, to improve lipid profile, and to control their body mass [26]. During aging, energy requirements along with energy intake are reduced, with an average reduction of 25% in energy intake between the age of 40 to 70. This reduction inevitably leads to weight and muscle loss, leading to negative effects in strength and body function [27]. Moreover, diet interventions seem to be important strategies, not only for the management of muscle mass and strength, but also in the prevention of diseases related to malnutrition. In the literature, there is evidence that diet interventions have a positive effect in supervising older adults and result in the better adoption of treatment strategies [28]. It should be noted that reduced energy intake and monotonous diets impair exercise adaptations related to muscle mass enhancement, mainly due to the reduction of energy intake and consequently due to reduced intake of important macro- and micro-nutrients that are important for the anabolic processes [29].

To the best of our knowledge, there are no studies in the literature in which sarcopenia and T2DM were treated simultaneously. Therefore, the aim of this review was to evaluate the diet and exercise interventions that have been applied for the treatment of sarcopenia and T2DM individually, and which exercise and diet interventions can be used simultaneously in order to effectively deal with the two disorders. The present review is mainly focused on studies which have focused on exercise treatment techniques (aerobic, resistance, or combined aerobic and resistance) as well as diet interventions that are known to best combat the two disorders. 

## 2. Methods

In the present review, we used studies using chronic exercise or/and diet intervention and clinical trials which included older adults suffering from sarcopenia or T2DM. We focused on treatment techniques that involved aerobic training, resistance training, or both trainings, as well as diet interventions or nutrition analysis. We also selected studies that used biomarkers of muscle mass and/or glycemic status.

An electronic search was performed using the following databases: PubMed and Google Scholar using the following search terms for each individual database:“resistance training” OR “resistance exercise” OR “strength training” OR “aerobic exercise’’“frailty” OR “muscle loss” OR “sarcopenia”“muscle hypertrophy” OR “muscle strength” OR “skeletal muscle mass”“diabetes mellitus” OR “type 2 diabetes mellitus OR “insulin resistance”“diet” OR “nutrition” OR “amino acids” OR “protein” OR “antioxidant” OR “life-style modification” OR “omega-3 supplementation” OR “vitamin D supplementation”“older adults” OR “elderly”

Recent literature published after 2000 was considered. We analyzed both titles and abstracts to decide whether a study could be included in our review. Articles were included if they fulfilled the following criteria: being peer-reviewed and/or research-based, met the search criteria, and being considered well-conducted studies. We also included seven reviews that were closest to our criteria. 

In order to assess the impact of the selected studies in sarcopenia we have included markers that are meaningful in the diagnosis of sarcopenia as stated from the various Consensus Group Statements [6,7,8,9,30]. In this context, sarcopenic markers should involve: muscle mass, muscle strength, and physical performance measurements (i.e., MWD, TUG, etc.). Similarly, we have included in the results (diabetic) the markers that have been related to the diagnosis of T2DM according to the ADA diagnosis criteria [21,31], such as blood pressure, cholesterol levels, triglyceride levels, insulin resistance, serum insulin levels, fasting plasma glucose, 2-h post-feeding glucose levels, HbA1c, and several hormones (i.e., glucagon, growth hormone, etc.). We also included IGF-1 in the diabetic markers [32]. 

## 3. Sarcopenia and T2DM

Sarcopenia and T2DM affect different systems of the human body, however, a bidirectional relationship between the two disorders has been gradually recognized. Sarcopenia is mainly characterized by type II muscle fiber atrophy, a reduced number of motor neurons and mitochondria, along with increased fat accumulation within the muscles [5,33]. Moreover, during the aging process, changes in hormonal levels contribute to sarcopenia. Indeed, lower levels of growth hormone, testosterone, thyroid hormones, and insulin-like growth factor may contribute to both muscle mass and strength loss [33]. Hormonal changes occur in both sexes indiscriminately and lead to a chronic inflammatory situation [33]. Regarding inflammation, while the aging process reveals an increase in inflammation markers, such as interleukin-6 (IL-6), tumor necrosis factor-alpha (TNF-α), and C-reactive protein (CRP), it is questionable as to whether this increase is due to the aging process or due to the diseases that accompany older adults [33]. Additionally, the function of muscle synthesis is restrained and muscles cannot be regenerated effectively, a situation known as anabolic resistance, while immobility seems also to be an important factor for muscle loss and vice versa [5].

Low protein intake and low levels of vitamin D are considered emerging nutritional imbalances among older adults, since they either both or separately are related to reduced muscle performance [33]. Generally, older adults consume 25% less food than they consumed during their adulthood, while the quality of food is also compromised. Data reveal that sarcopenia can be affected by a genetic predisposition that relates to the structure, function, and metabolism of skeletal muscle fibers [18]. The level of muscle mass and strength that can be achieved in older adults is not only related to muscle tissue loss but is also related to the level of muscle performance that has been achieved during adulthood [18]. In summary, the main factors that lead to sarcopenia include low physical activity levels, low energy intake, progressive fibrosis increase, dysfunction of muscle metabolism, chronic inflammation, oxidative stress, and neuromuscular degeneration (Figure 1). As a result of the above factors, the human body gradually loses muscle fibers by the age of fifty, while by the age of eighty, 50% of skeletal muscle tissue has been lost. This loss rate is also observed in older adults who were athletes as adults [34].

The main factors that contribute to the prevalence of diabetes are genetic, increased body weight, and an unhealthy lifestyle. The consumed food is converted to glucose, through digestion, and then is diffused in the blood stream. Through glycolysis, glucose provides the energy that the body needs in order to cope with its daily activities. Insulin is a hormone secreted by the pancreas and allows glucose to be absorbed by cells. In people suffering from diabetes mellitus, the quantity of secreted insulin is never enough in order to induce sufficient glucose absorption, leading to the continuous circulation of glucose in the blood stream. One reason for this limited absorption is insulin resistance in skeletal muscle tissues. In this case, these tissues cannot detect the circulating blood stream glucose, cells starve, and the pancreas is then pressed to increase insulin production. Contrary to the skeletal muscle, which lacks glucose, other human organs and tissues are exposed to unhealthily high glucose levels (e.g., refinery, kidneys, brain, cardiovascular, central neural system) [35,36]. Another reason for insufficient glucose absorption is the dysfunction of the β-cells of the pancreas, in which case the produced insulin cannot overcome the demand of skeletal tissues [35,36]. It has been suggested that the insulin resistance is the primary cause preceding the later dysfunction in β-cells’ production [37].

Diabetes mellitus and sarcopenia share common causes (e.g., reduced physical activity and low levels of anabolic hormones, such as insulin growth factor-1, testosterone, and ghrelin), while in diabetic patients the intramuscular fat storage is up to three times greater when compared to healthy older adults [36,38,39]. Insulin resistance in skeletal muscle is the most important factor that amplifies sarcopenia in T2DM patients. Proteins and amino acids are the main macronutrients that preserve muscle tissue. Protein synthesis takes place mostly in the postprandial state, when the concentrations of insulin and amino acids reach their maximum levels [40,41]. Specifically, insulin function promotes protein synthesis in skeletal muscle cells only when there is an increased availability of intramuscular amino acids, while a lack of insulin leads to protein breakdown [40,41]. It is clear that low insulin levels may lead to insufficient protein synthesis and increased protein degradation, which in turn may lead to sarcopenia [42]. Consecutive muscle loss provokes metabolic changes, which gradually amplifies hyperglycemia and chronic inflammation. Of importance, hyperglycemia relates to metabolic abnormalities that can also lead to muscle wasting [20,43]. The anabolic effect of insulin on muscle mass in combination with adequate amino acids intake could be an effective countermeasure for muscle loss, not only for people who suffer from diabetic mellitus, but generally for older adults [44,45]. In Figure 2, a generic view of sarcopenia and diabetes interactions can be seen.

Relevant studies highlight this bidirectional relationship, which in turn accelerates the progress of both sarcopenia and T2DM, while it is strongly suggested that clinical examinations should involve both conditions [45,46]. Multiple studies provide evidence that sarcopenia relates to diabetes mellitus and vice versa (Table 1). From the included studies, it can be easily observed that patients with sarcopenia have a high prevalence of T2DM [24,47,48] and vice versa [23,47,48,49,50,51]. It is also evident that low-value muscle mass markers are strongly related to reduced glycemic control [38,52,53].

## 4. Exercise and Diet to Control Sarcopenia and Type 2 Diabetes Mellitus in Older Adults

Exercise of the aerobic component mainly improves metabolic regulation and cardiovascular function, while aerobic exercise is known to induce positive effects in combating sarcopenia and T2DM [55]. An improved glycemic profile is observed through reductions in glycosylated hemoglobin (HbA1c) [56], fasting plasma glucose [51], and insulin resistance [57]. Moreover, high-intensity interval training is very effective and has the advantage of being very time-efficient [55,56,58]. Aerobic exercise has also been found to improve skeletal muscle mass, especially leg strength [56,59,60] and this is the reason why it is highly recommended in older adults [61].

Resistance training, on the other hand, is known to increase muscle mass, muscle strength and physical performance [57,60,62,63]. Indeed, resistance training is especially appropriate to older frail adults [59,64,65]. With resistance training, leg strength is improved [64,65,66,67] and subjects perform much better in physical performance [60,64,65]. Of importance, resistance exercise has also been found to exert beneficial effects on glycemic profile and insulin sensitivity [68,69]. Another study noted that while HbA1c did not change significantly [67,70], a great response was observed in diabetes medication [70]. Resistance training has also been found to greatly elevate IGF-1 [59], which in turn affects insulin sensitivity [71]. IGF-1 secretion is also known to be greatly associated with muscle generation, muscle mass, and muscle strength [59]. Of interest, one study noted that no significant results have been observed on older adults while, exercise was found to increase the risk of musculoskeletal injury [72].

Many studies combining aerobic and resistance training have been used in order to maximize the benefits of both types of exercise. In particular, a combined aerobic and resistance regime was found to improve glucose regulation by reducing fasting glucose [73], levels of fructosamine [63], or triglycerides [68], while it caused increases in serum IGF-1 levels [59] and in insulin sensitivity [68]. Sarcopenic individuals should also be benefited by using combined training, since it improves muscle mass [59], strength [59,73,74], and physical performance [73], despite the fact that combined exercise is not as effective as pure resistance training to cause improvements in skeletal muscle [59]. 

In older adults, an adequate protein intake could be a successful strategy for combating sarcopenia. Indeed, dietary proteins provide amino acids, which are essential for the synthesis of muscle protein [66,74,75] and glucose control, and result in reduced fasting glucose [66], reduced levels of triglycerides [66], and even improved insulin responses [76]. Improvements in physical performance have also been observed [74]. There is no significant association between diabetes and total dairy intake, although a high intake of milk may reduce T2DM risk among men [77]. In some studies, low levels of vitamin D have been observed to be related with muscle mass loss and sarcopenia [78,79]. However, the infusion of vitamin D in frail older patients did not show any significant result [72]. Vitamin D is also associated with an elevated risk of future diabetes in older people [80]. Vitamin B12 might also play a role in the muscle mass loss, however, this relationship is not established [72]. Omega-3 fatty acid intake was negatively associated with the presence of sarcopenia [81]. Sarcopenia seems also to be related to the concentration of carotenes, since higher plasma concentrations were associated with a reduced risk of low grip, hip, and knee strength [82]. 

Finally, it seems that a combination of exercise and diet has positive effects in improving muscle mass, muscle strength and physical performance [62,63,83]. 

Table 2 summarizes the reviewed studies. The first four examined both markers: sarcopenia and T2DM. The next eight studies examined only sarcopenic markers, and the last seven studies included only T2DM markers.

## 5. Discussion and Conclusions

Sarcopenia and T2DM seem to coexist among patients suffering of either disorder, since it is not only the progression of diabetes mellitus that may lead to muscle mass loss, but also the progression of sarcopenia may lead to a diabetic condition [23,24]. This means that studies and compensation techniques should include measurements and clinical examinations that involve both disorders. Aerobic and resistance exercise and diet independently have a significant positive impact on each of the diseases. Aerobic and resistance exercise are the most well-studied types of training and therefore the most preferable for older individuals with health complications. Results have shown that all types of exercise greatly contribute to both disorders. In particular, aerobic exercise improved the glycemic profile through reductions in glycosylated hemoglobin (HbA1c) [56], fasting plasma glucose [56], and insulin resistance [68]. Studies in a healthy population have also revealed a positive impact on insulin resistance [68]. This means that aerobic exercise has a clear beneficial effect on T2DM patients, since they exercise under supervision [86,87]. Aerobic exercise also improves skeletal muscle mass, especially leg strength [51,55] and is recommended for the older adults [61], although it is more efficient when combined with resistance training [59,61,73]. Aerobic exercise contributes to the mitochondrial biogenesis and restoration of mitochondrial metabolism, which in turn reduces the expressions of catabolic genes and increases muscle protein synthesis, enhancing by this way muscle hypertrophy [73]. High-intensity interval training is very effective and has the added benefit of being very time-efficient [55,56]. Apart from those, aerobic exercise greatly improves aerobic capacity to prior sedentary subjects and even to older adults with cardiovascular disease [25]. Specifically, regular aerobic exercise decreases aortic stiffness and enhances blood flow, resulting in the better control of vascular circulation, whilst increases in HDL-C improve the atherogenic profile and reduce risk factors for cardiovascular events [25]. Moreover, a lower score in depression is also experienced by exercising subjects [25], which is very important to improve the quality of life. 

Resistance training, on the other hand, is mainly known to increase muscle mass, muscle strength, and physical performance. This makes resistance training especially appropriate to older, frail adults. Improvements in muscle quality [59,64,65] and strength [64,65,66] along with the increased skeletal mass reduce the risks of falls and improve physical performance [57,64,65], making resistance exercise an appropriate physical activity for sarcopenic individuals [86]. However, participation in resistance exercise programs definitely needs supervision since this type of exercise increases the risk of musculoskeletal injury in older adults [72]. Moreover, resistance training reduces intramuscular fat and improves insulin sensitivity [68,86], although aerobic exercise has a better impact on metabolic markers [25]. Resistance training also greatly elevates IGF-1 [59], which in turn affects insulin sensitivity [71]. IGF-1 secretion is also known to be greatly associated with muscle generation, muscle mass, and muscle strength [59]. Of importance, resistance exercise has also been found to exert beneficial effects on health-related parameters such as bone mineral density, blood pressure, lipid profiles, and cardiovascular health [55,88]. It should be stressed that during physical activity, a strict blood pressure control is required for the avoidance of diabetic retinopathy or nephropathy [89]. A reduction in obesity is another important factor for glycemic profile improvement, since reductions in intramuscular fat can improve insulin sensitivity [59]. However, caution should be taken during resistance exercise since it may increase the risk of injury, while also leading to dropouts due to the unvaried nature of exercises [86].

In the literature, many studies have used combined aerobic and resistance training in order to maximize the benefits of both types of exercise. In particular, a combined aerobic and resistance training program along with improvements in cardiovascular function and metabolic factors not only achieves better glycemic control, but also improves muscle mass, strength, and function [59,73,84,86,90,91]. However, it should be noted that some studies did not observe an improvement in HbA1c [84]. Since both types of training contribute to health improvement, the American Diabetes Association and the Japan Diabetes Society recommend daily participation in activities that include both the aerobic and resistance type of exercise [92,93]. Nonetheless, despite the fact that there is no specific health factor to avoid exercise, older adults who suffer from diabetes mellitus do not participate in organized exercise programs due to a lack of time, an inability to control glucose during exercise, denial in recognizing obesity as a health problem, a lack of appropriate infrastructure, and other social factors [89,94].

Nutritional interventions/therapies in people who suffer from diabetes mellitus aim to achieve healthy levels of glucose, lipid profile, and blood pressure. Energy intake restriction is an effective strategy for obese or overweight individuals in order to control their body mass, while it is critical for the avoidance of metabolic syndrome [59]. However, while fat loss may be an essential parameter for older adults suffering from T2DM, it may also lead to muscle mass loss and cause the exacerbation of sarcopenia. The observed skeletal muscle loss during energy restriction could be a result of inadequate protein intake [95,96]. This means that this strategy should be avoided, especially in older adults who suffer from endogenous muscle loss [97].

In older adults, an adequate protein intake could be a successful strategy for combating sarcopenia. Regarding dietary proteins, two main research areas have been developed. The first deals with the effect of protein intake on weight loss and glycemic control and the second deals with the long-standing interest of the preservation of muscle mass [98]. Indeed, dietary proteins provide amino acids that are essential for the synthesis of muscle protein [66,75] and glucose control and which result in reduced fasting glucose [66], triglycerides [66], and improved insulin responses [76], independently of age [93,94]. Specifically, branch-chain amino acids (i.e., leucine, isoleucine, and valine) regulate the release of hormones (for example, leptin, GLP-1, and ghrelin) that can potentially affect food intake and glycemia levels [87,99]. Amino acids and insulin are anabolic signals that activate the mammalian target of rapamycin complex 1 (mTORC1) and protein kinase, altering the growth of energy-consuming tissues and decreasing protein breakdown [87]. It is reported in the literature that a mixture of amino acids enriched with leucine may compensate for the reduced muscle synthesis in older adults, since leucine can increase anabolic response [33,100]. B-hydroxy-β-methylbutyrate (HMB) is a basic metabolite of leucine that contributes to protein synthesis; however, it is found to be reduced during aging [100]. Undeniably, in older adults HBM supplementation caused reductions in muscle mass degradation was and found to improve muscle strength [101,102].

Low levels of vitamin D have been observed to be related with muscle mass loss and enhanced sarcopenia [29,78,79,103]. Specifically, low levels of vitamin D are also related to reduced glycemic control, since vitamin D receptors located inside the pancreas and skeletal muscle contribute to glucose and muscle homeostasis [29,33,104,105]. However, due to equivocal results regarding the effects of vitamin D on physical strength and performance, more studies are needed to determine the exact role of vitamin D in sarcopenia [29,72]. Vitamin B12 might also play a role in the muscle mass loss, however, this relationship is not established [72]. There is no significant association between diabetes and total dairy intake, although a high intake of milk may reduce T2DM risk among men [77].

Sarcopenia is increasingly recognized as an inflammatory condition caused by cytokines and oxidative stress [29]. Specifically, polyunsaturated fatty acids have found to be strong anti-inflammatory agents and therefore they may improve muscle function [81,103]. This is also holds true in older adults, since it has been found that increased strength occurs after consumption of fishes, however more studies are needed to further support this finding [29,103]. Cooperatively, oxidative stress can damage biomolecules, such as DNA, lipids, and proteins, since reactive oxygen species (ROS) are present in every cell compartment. Reactive oxygen species are normally compensated for by antioxidant defense mechanisms, including the enzymes peroxide dismutase and glutathione peroxidase, as well as exogenous dietary antioxidants such as selenium, carotenoids, tocopherols, flavonoids, and other polyphenols [106,107,108]. In older adults, excessive ROS concentrations may lead to oxidative stress and result in the loss of muscle mass and strength [29]. Generally, there is a limited number of studies examining the effects of antioxidant supplements on muscle strength, and therefore the results are debatable [82,109].

Undoubtedly older adults need much more dietary proteins than younger individuals in order to support their wellness, recovery from injuries, and to maintain their independence in everyday life activities. Older adults must compensate for changes related to protein metabolism, while at the same time they need more protein to counteract inflammation and catabolism caused by acute and chronic diseases [41]. The existing guidelines for dietary proteins suggest 0.8 g of protein per kilogram of body mass per day (gr/kg BM/d) independently of sex or age [42]. Unfortunately, this is a rather general recommendation and does not account for changes caused by the aging process (e.g., metabolism, immunity, hormonal levels, and progressive frailty). Based on data of recent investigations, it has been suggested that lean body mass can be better preserved if older adults consume larger quantities of dietary proteins than the general population [57]. In the PROT-AGE project [57] it was proposed that an average daily protein intake of 1–1.2 gr/kg BM/d (or approximately 25–30 gr/meal) be consumed by healthy older adults in order to preserve their muscle mass. For the older adults suffering from acute or chronic diseases, in the PROT-AGE project it is proposed that an increased protein intake (i.e., 1.2–1.5 gr/Kg BM/d) should be dependent on the characteristics of the disease (e.g., the severity of the disease, the patient’s dietary condition before disease, and the impact of disease on patient’s diet). Of importance, it has been suggested that a protein consumption of up to 2 gr/kg BM/d be adhered to for older adults suffering from severe disease/injury/starvation [57].

To the best of our knowledge, studies that combine both exercise and nutritional interventions are mainly focused on obese people [110,111,112,113]. Apart from that, there exists very limited research on proposing methods to simultaneously compensate for both diseases. In a relevant study, diabetic mice were supplemented with a ginseng diet and followed aerobic exercise, and were found to have improvements in muscle mass and on insulin resistance [114]. Generally, while there is an agreement that both diseases should be treated simultaneously, researchers seem to ignore people suffering from both diseases and prefer to examine the treatment techniques separately. Nevertheless, sarcopenia and diabetes mellitus have high prevalence among the older adults, which in turn may lead to a low quality of life. The available data show that a combination of supervised combined training along with protein intake > 1 gr/kg/d seems to be the safest treatment, if not the best, in order to compensate both sarcopenia and T2DM. Of note, nutrition counseling is also important to achieve diet adherence [28,115]. Due to the fact that one disorder fuels the other, more studies are needed involving people suffering both disorders. Since sarcopenia and T2DM have been widespread among the population in the modern era, it is imperative to find the best combination of exercise and diet interventions that will be succeed in managing sarcopenia and diabetes mellitus.

## Figures and Tables

**Figure 1 jfmk-07-00048-f001:**
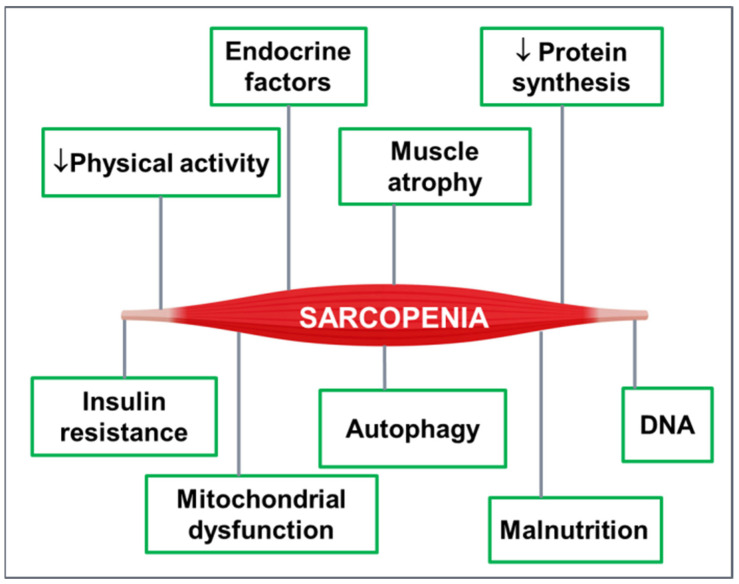
Pathophysiological factors of sarcopenia.

**Figure 2 jfmk-07-00048-f002:**
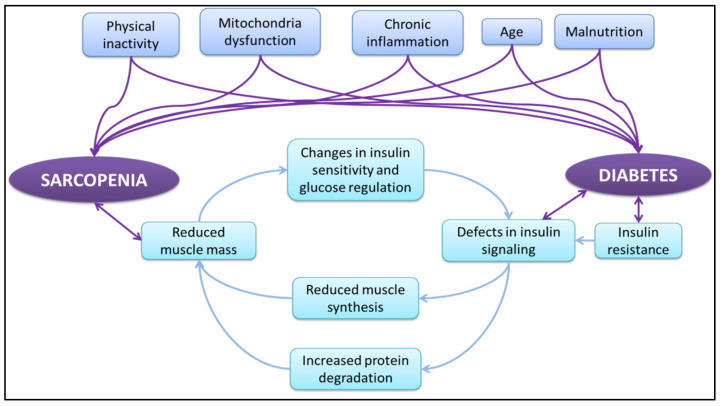
Interaction of sarcopenia and diabetes.

**Table 1 jfmk-07-00048-t001:** Studies establishing that sarcopenia and diabetes mellitus are closely related.

Study	Muscle Mass Criteria	Study Design	Result
Kim et al., 2010 [23]	SMI < 2 SDs	Cross-sectional study 2 groups: -T2DM: *n* = 414 -control group: *n* = 396	SMI values were significantly decreased in patients with diabetes compared with control group Patients with diabetes had three times higher risk of having a low SMI than control group
Wang et al., 2016 [49]	AWGS criteria	Cross-sectional study Community-dwelling Chinese citizens (≥60 years) 2 groups: -T2DM Patients: *n* = 116 men + 120 women -control group: *n* = 404 men + 450 women	The prevalence of sarcopenia and pre-sarcopenia was significantly higher in diabetic patients than in healthy controls Diabetic patients was associated with a 1.56-fold increased risk of sarcopenia using AWGS criteria
Kim et al., 2014 [50]	Three different formulas for skeletal mass: (a) ASM/height2, (b) ASM/weight, (c) TSM/weight	Cross-sectional study Adults (≥65 years) 2 groups: -T2DM: *n* = 59 men + 85 women -control: *n* = 130 men + 140 women	Older men with T2DM showed significantly lower ASM than those without diabetes The risk of low muscle mass (in all formulas) was significantly higher in older men with T2DM
Souza et al., 2019 [24]	EWGSOP criteria	Cross-sectional study Older adults (>60 years) *n* = 1078	Diabetes mellitus was present in 36.87% of the patients with sarcopenia using EWGSOP criteria
Leenders et al., 2013 [53]	DEXA, sit-to-stand test, handgrip test	Cross-sectional study (3 months) Community dwelling and still living Independently older men 2 groups: -T2DM: *n* = 60 men -control group: *n* = 32 men	Leg lean mass and ASM were significantly lower in older men with T2DM compared with normoglycemic controls Leg extension strength was significantly lower in the group with T2DM Significant longer sit-to-stand time for group with T2DM compared with normoglycemic group Significant lower handgrip strength for group with T2DM when compared with normoglycemic controls.
Kalyani et al., 2015 [52]	Knee extensor strength divided by DEXA-derived leg lean body mass	Longitudinal study (7.5 years) Adults (25–96 years) *n* = 984	Muscle strength (knee extensor strength) and muscle quality (knee extensor strength/leg lean mass) were all significantly decreased from lower to higher HbA1c. Hyperglycemia is associated with persistently lower muscle strength with aging
Park et al., 2009 [38]	DEXA	Longitudinal study (6 years) well-functioning community-dwelling Adults (70–79 years) *n* = 2675	Older adults with either diagnosed or undiagnosed Type 2 diabetes showed excessive loss of appendicular lean mass and trunk fat mass compared with nondiabetic subjects Thigh muscle cross-sectional area declined two times faster in older women with diabetes than their nondiabetic counterparts
Anagnostis et al., 2020 [47]	Multiple criteria depending on studies (EWGSOP, AWGS, FNIH)	Systematic review Patients with T2DM or sarcopenia *n* = 1832 + 1159	Patients with T2DM demonstrated a higher risk of sarcopenia (using EWGSOP or AWGS or FNIH criteria) compared with euglycemic subjects Patients with T2DM have an increased risk of sarcopenia (using EWGSOP or AWGS or FNIH criteria) compared with euglycemic subjects
Ai et al., 2021 [51]	Multiple criteria depending on studies (EWGSOP, AWGS, FNIH, LMM, LMS, LSMI)	Systematic review Patients with T2DM *n* = 16634	The pooled prevalence of sarcopenia in patients with T2DM was 18% Elder age, male gender and chronic hyperglycemia, Osteoporosis were significant risk factors for Sarcopenia
Veronese et al., 2019 [48]	AWGS, EWGSOP criteria	Systematic review Adults with mean age = 65.4 years *n* = 54676	Diabetic participants had an increased prevalence of sarcopenia (using EWGSOP or AWGS criteria) compared to controls Sarcopenia (using EWGSOP or AWGS criteria) was associated with an increased odds of having diabetes
Chung et al., 2019 [54]	AWGS criteria	Systematic review Asian aged ≥60 years 2 groups: -diabetics: *n* = 1537 -non- diabetics: *n* = 5485	Diabetics showed a significantly higher risk of sarcopenia (using AWGS criteria) than non-diabetics

**Table 2 jfmk-07-00048-t002:** Treatments for people with diabetes mellitus or sarcopenia and results in T2DM/sarcopenic markers.

Study	Study Design	Duration	Exercise Intervention	Nutritional Intervention	Sarcopenic Markers	T2DM Markers
Mitranun et al., 2014 [56]	Parallel-group randomized trial Diabetic patients (50–70 years) *n* = 16 men + 29 women 2 groups: continuous AT, interval AT	12 weeks	3 sessions/week AT: 30–40 min walking		Significant increase in leg muscle strength for both AT No significant change in upper body strength	Significant decrease in fasting glucose concentration and insulin resistance in both exercise groups Significant decrease in HbA1c in interval AT
Tan et al., 2015 [73]	Randomized control trial Diabetic patients (>60 years) *n* = 13 men + 12 women 2 groups: CT, CON	6-month	3 sessions/week CT: 30 min moderate AT; 10 min RT		Significant increase in the leg muscle strength for CT group Significant increase in 6-MWD for the CT group	Significant decrease for CT group in concentrations of fasting and 2-h post-glucose challenge plasma glucose, serum insulin, HbA1c
Egger et al., 2013 [84]	Parallel-group randomized trial Diabetic patients (64.8 ± 7.8 years) *n* = 13 men and 19 women 2 groups: CON, Hypertrophy CT, Endurance CT	8 weeks	7 sessions/week Hypertrophy CT: RT:10–12 repetitions,70% 1-RM; AT 1 h/day Endurance CT: RT: 25–30 repetitions, 40% 1-RM; AT 1 h/day		Significant increase in muscle strength and muscle mass for both groups Significant increase in strength in hypertrophy CT versus endurance CT	HbA1c did not change significanlty Significant reductions in fasting glucose, fructosamine
Chen, et al., 2017 [59]	Randomized control trial Sarcopenic patients with obesity (>65 years) *n* = 60 4 groups: AT, RT, CT, CON	12 weeks	RT: 60–70% 1-RM, 3 sets 8–10 repetitions, 2 sessions/week AT: 1 h, 2 sessions/week CT: 1 session/week RT, 1 session/week AT		Significant increase in skeletal muscle mass and strength for AT, RT, CT groups vs. CON Significant increase in grip strength at weeks 8 and 12 in the RT group vs. AT, CT, CON groups	Significant increase in IGF-1 concentration at week 8 for the RT, CT groups vs. AT, CON groups. At Week 12, no significant differences were observed among the four groups.
Lustosa et al. 2011 [64]	Randomized, crossover trial Pre-frail community-dwelling women (>65) *n* = 322 groups: RT, CON	10 weeks	3 sessions/week RT 75% 1-RM, 8 repetitions		Significant improvement in TUG, 10-MWT and knee extensor’s muscle strength	Not measured
Latham et al., 2013 [72]	Randomized control trial Frail patients (>65 years) *n* = 243 4 groups: RT, CON, Vitamin D, placebo	10 weeks	RT 60–80% 1-RM	Single dose of vitamin D	No significant results in TUG and MWD tests Exercise non-significantly increased the risk of musculoskeletal injury	Not measured
Seynnes et al., 2004 [65]	Randomized control trial Frail patients (>70 years) *n* = 22 3 groups: high intensity RT, low intensity RT, CON	10 weeks	3 sessions/week Highly intensive RT: 80% 1-RM, 3 sets of 8 repetitions Low intensive RT: 40% 1-RM, 3 sets of 8 repetitions		RT groups significantly improved knee extensor strength, endurance, stair-climbing power, and chair-rising time 6-MWD significantly improved only in the high intensity RT	Not measured
Semba et al., 2003 [82]	Cross-sectional study Non-disabled to severely disabled women (>65 years) *n* = 669	8 years		Data analysis	Higher plasma concentrations of α-carotene, β-carotene, β-cryptoxanthin, and lutein/zeaxanthin were associated with reduced grip, hip, and knee strength with 95% confidence interval	Not measured
Bischoff, et al., 1999 [79]	Cross-sectional study Frail patients (>65 years) *n* = 216 men + 103 women	2 years		Data analysis	Vitamin D related significantly to both sexes with the strength degradation	Not measured
Okamura et al., 2020 [81]	Cohort study Diabetic people (>65) *n* = 180 men + 162 women	2 years		Data analysis	Low omega-3 fatty acids levels intake was significantly associated with the presence of sarcopenia based on Japan Society of Hepatology	Not measured
Takahashi et al., 2021 [78]	Cohort study Diabetic people (>65) *n* = 112 men + 85 women	2 years		Data analysis	Low levels of vitamin D, intake was significantly related to the loss of muscle mass Low levels of vitamin B1 and vitamin B12 intake was non-significantly related to the loss of muscle mass Vitamin A, vitamin B1, vitamin B2, vitamin B6, vitamin C and vitamin E were not found to be significantly related to the loss of muscle mass	Not measured
Tieland et al., 2012 [75]	Randomized, Double-Blind, Placebo-Controlled Trial Frail patients (>65 years) *n* = 65	24 weeks		15 g protein 2 times/day	Skeletal muscle mass did not change significantly in the protein or placebo group Leg extension strength significantly increased in protein group Physical performance improved significantly in protein group TUG, MWD and handgrip strength were not significantly associated with protein or placebo group	Not measured
Terada et al., 2013 [58]	Parallel-group randomized Diabetic patients (55–75 years) *n* = 15 = 8 males + 7 women 2 groups: interval AT, continuous AT	12 weeks	5 sessions/week AT high intensity interval exercise: cycling and treadmill walking; 1-min intervals at 100% VO2R followed by 3-min recovery intervals at 20% VO2R; AT moderate intensity continuous exercise: stationary cycling and treadmill walking; continuous exercise at 40% VO2R;		Not measured	HbA1c, fasting blood glucose did not significantly change from baseline
Amamou et al., 2017 [85]	Parallel-group randomized trial Overweight adults (60 -75 years) with at least 2 factors of metabolic syndrome *n* = 17 men + 14 women 2 groups: high protein, high protein + RT	6-weeks	3 sessions/week RT: 65–80% 1-RM, 2 sets of 8–15 repetitions	Caloric intake was reduced by 500 kcal/d in all participants and protein intake ~1.4 g/kg/d	Not measured	Significant decrease in fasting glucose and triglycerides in both groups
Kadoglou et al., 2013 [68]	Randomized control trial Diabetic people (>55) *n* = 100 4 groups: AT, RT, CT, CON	16 weeks	AT: 60–75% of maximum heart rate RT: 60–80% 1-RM CT: above AT + RT		Not measured	All exercise groups significantly ameliorated glycemic profile, since the reduction in fasting plasma glucose, HbA1c, fasting insulin levels, insulin resistance and triglycerides levels was significant compared with the control group
Castaneda et al., 2002 [70]	Randomized control trial Diabetic people (>55) *n* = 62 2 groups: RT, CON		3 sessions/week RT: 60–70% 1-RM		Not measured	Significant decrease in HbA1c, fasting plasma glucose concentrations for RT group Non-significant decrease in serum triglyceride concentrations in RT group
Van Loon et al., 2003 [76]	Cross-sectional study T2DM + healthy people (>55) *n* = 20 10 were T2DM patients and 10 were CON	1 week		2 sessions Session 1: both groups received carbohydrates Session 2: both groups received carbohydrates with amino acid/protein mixture	Not measured	Significant increase in insulin responses for mixture intervention group Non-significant differences were observed in plasma glucose, glucagon, growth hormone, IGF-I, within 2-h time frame for the intervention groups.
Huang et al., 2014 [66]	Cross-sectional study Diabetic people (>65) *n* = 210 4 groups: low protein (<0.6 gr/d/kg), moderate protein (>0.6, <0.8 gr/d/kg), high protein (>0.8, <1 gr/d/kg), very high protein (>1 gr/d/kg)	unclear		Data analysis	Not measured	Significant reduction in HbA1c and triglycerides in high and very high protein groups There were observed non-significant reductions in fasting plasma glucose
Moslehi et al., 2015 [77]	Nested case–control study Healthy people that developed T2DM *n* = 698 2 groups: T2DM, CON	7 years		Data analysis	Not measured	Milk intake decreased non-significantly the T2DM risk in men but not in women. There was no significant association between diabetes and total dairy intake No significant association was observed between diabetes and carbohydrate, protein, fermented dairy, grain, fruit, vegetable, meat
Vlietstra et al., 2018 [60]	Systematic review Adults with sarcopenia (>60 years)	3–6 months	Multiple exercise interventions		Knee-extension strength, TUG, appendicular muscle mass and leg muscle mass significantly improved in response to exercise interventions MWD did not significantly improved	Not measured
Yoshimura et al., 2017 [62]	Systematic Review Adults with sarcopenia(>60 years)	3–6 months	Multiple exercise interventions	Multiple nutritional interventions	Exercise interventions did not significantly change muscle mass, muscle strength, and walking speed Nutritional interventions did not significantly change muscle strength A combined intervention of exercise and nutrition did not significantly change the walking speed.	Not measured
Wu et al., 2021 [63]	Systematic Review older adults with sarcopenia (>65 years)		RT	Multiple nutritional interventions	RT alone and the combination of RT and nutrition significantly increased handgrip strength and improved dynamic balance	Not measured
Liao et al., 2019 [83]	Systematic Review Older adults with a high risk of sarcopenia or frailty and physical limitations (>60 years)		RT or a multicomponent exercise regime that consisted of MSE, aerobic exercise, balance training, and physical activity training	Protein supplementation	The protein supplement + exercise group exhibited significant improvements in the whole-body lean mass, appendicular lean mass, leg strength, and walking capability	Not measured
Lucato et al., 2017 [80]	Systematic review Adults evolved T2DM (>60 years) *n* = 28258			Vitamin D analysis	Not measured	Hypovitaminosis D is significantly associated with an elevated risk of future diabetes in older people
Hovanec et al., 2012 [69]	Systematic review Adults with T2DM (>65 years)		RT		RT had increased significantly lower body muscle strength, upper body muscle strength and whole body muscle strength	RT did not cause any significant decrease in HbA1c and fasting glucose
Irvine et al., 2009 [67]	Systematic review Adults with T2DM (mean age = 58.4) *n* = 372		Progressive RT		Progressive RT resulted in significant improvements in strength when compared to AT or no exercise	Compared to control, progressive RT led to small and statistically significant absolute reductions in HbA1c of 0.3% Compared to AT there were no significant differences in HbA1c

## Data Availability

Not applicable.

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
