# Peer review of "Exercise and Nutrition Strategies for Combating Sarcopenia and Type 2 Diabetes Mellitus in Older Adults"

_jfmk, 2022, doi:10.3390/jfmk7020048_

Round 1

Reviewer 1 Report

Dear Authors.

My comments were fully addressed.

The quality of the manuscript in its actual form has increased after these last changes.

Author Response

My comments were fully addressed. The quality of the manuscript in its actual form has increased after these last changes.

Reply

The authors would like to thank the Reviewer for his/her constructive criticism and valuable comments to improve the manuscript.

Reviewer 2 Report

This is a literature review summarizing the relationship between sarcopenia and type 2 diabetes (T2DM) and studies of diet and exercise interventions for both sarcopenia and T2DM. This is an important topic given the critical role of muscle metabolism on both conditions, and there are many outstanding questions related to impact of current interventions (resistance training, dietary protein, amino acids, HMB, vitamin D) in this population.  

While the authors are comprehensive in their review, there are important limitations in their presentation of studies that must be addressed.  Specific comments are below.

1. in Table 1, the definitions for Sarcopenia should align with (one or
more) of the Consensus Group Statements, or clearly state otherwise.
For example, in the first reference, Kim et al., 2010 used SMI values
alone, which were significantly decreased in patients with diabetes
compared with control group.  This would not support the statement "
Patients with diabetes had three times higher risk of sarcopenia than
control group" using e.g., EWGSOP criteria.  Please be consistent here.

2.  Standardize descriptions in Table 2 under "Sarcopenic Markers:" for
example, current descriptions vary, e.g., "Leg muscle strength for both
AT increased," to "Significant change in peak power output for AT
interval," to " Great gain in strength for both groups before and after
intervention" to "Superior strength gain in hypertrophy CT ."   Is there
a scientific rationale for the terms "Increased" vs "Significant change"
vs "Great gain" vs "Superior gain"?  If not, please find another way to
communicate these outcomes to reader in a more objective manner. 

3.  Please note:  this comment in #2 also applies to descriptors for "T2DM markers:"
example:  "Insulin responses were dramatically increased..." vs "All
active groups significantly ameliorated glycaemic profile..."

4.  Separate and standardize descriptions in Table 2 under "Sarcopenic
Markers." For example, current descriptors include all variables from
"leg muscle strength" to "peak power output" to "skeletal muscle" to
"serum IGF and vitamin D."  This makes it difficult to identify any
consistency.   

5.  Please note:  this comment in #4 also applies to descriptors
for "T2DM markers," for example, "trunk and body fat mass" and "HBA1C"
and "serum IGF1" are all included under T2DM markers; the rationale is
unknown.

6.  Also, in Table 2, with a few exceptions, there are no comments on
"Sarcopenic markers" that were measured but which did not change
significantly.  This should also be noted. "

Author Response

The authors would like to thank the Reviewer for his/her constructive criticism and valuable comments to improve the manuscript.

This is a literature review summarizing the relationship between sarcopenia and type 2 diabetes (T2DM) and studies of diet and exercise interventions for both sarcopenia and T2DM. This is an important topic given the critical role of muscle metabolism on both conditions, and there are many outstanding questions related to impact of current interventions (resistance training, dietary protein, amino acids, HMB, vitamin D) in this population. 

While the authors are comprehensive in their review, there are important limitations in their presentation of studies that must be addressed.  Specific comments are below.

1. in Table 1, the definitions for Sarcopenia should align with (one or more) of the Consensus Group Statements, or clearly state otherwise. For example, in the first reference, Kim et al., 2010 used SMI values alone, which were significantly decreased in patients with diabetes compared with control group.  This would not support the statement "Patients with diabetes had three times higher risk of sarcopenia than control group" using e.g., EWGSOP criteria.  Please be consistent here.

Reply

We would like to thank the reviewer for the comment. In the revised manuscript, we have removed from the Table1 the studies that did not follow the criteria of the Consensus Group Statements (Table 1).

2.  Standardize descriptions in Table 2 under "Sarcopenic Markers:" for example, current descriptions vary, e.g., "Leg muscle strength for both AT increased," to "Significant change in peak power output for AT interval," to " Great gain in strength for both groups before and after intervention" to "Superior strength gain in hypertrophy CT."   Is there a scientific rationale for the terms "Increased" vs "Significant change" vs "Great gain" vs "Superior gain"?  If not, please find another way to communicate these outcomes to reader in a more objective manner.

Reply

In the Table 2 of the revised manuscript, we have corrected the section “Sarcopenic markers”, in a way that the results of each paper are now described with the statistical terms of “significant” and “not significant” (Table 2, “Sarcopenic markers” column).

3.  Please note:  this comment in #2 also applies to descriptors for "T2DM markers:" example: "Insulin responses were dramatically increased..." vs "All active groups significantly ameliorated glycaemic profile..."

Reply

In the Table 2 of the revised manuscript, we have corrected the section “T2DM markers”, in a way that the results of each paper are now described with the statistical terms of “significant” and “not significant” (Table 2, Table 2, “Sarcopenic markers” column).

4.  Separate and standardize descriptions in Table 2 under "Sarcopenic Markers." For example, current descriptors include all variables from "leg muscle strength" to "peak power output" to "skeletal muscle" to "serum IGF and vitamin D."  This makes it difficult to identify any consistency.  

Reply

In the Table 2 (“Sarcopenic markers” column) of the revised manuscript, we have tried to present the included studies in a way that there will be a consistency regarding the presentation of the measured variables beginning with the performance parameters and continue with the nutritional parameters. However, this was somehow difficult because we had to have in mind and the next comment (#5) where the presentation of the next column (T2DM) should also be in a specific order (Table 2, “Sarcopenic markers” column).

Moreover, in the Methods section of the revised manuscript, in a new paragraph it is clearly stated the variables which are used for the diagnosis of sarcopenia (Methods section, par. 4, lines 125-133).

The relevant text now reads:

“In order to assess the impact of the selected studies in sarcopenia we have included markers that are meaningful in the diagnosis of sarcopenia as stated from the various Consensus Group Statements [6-9, 114]. In this context, sarcopenic markers should involve: muscle mass, muscle strength, physical performance measurements (i.e., MWD, TUG, etc.). Similarly, we have included in the results (diabetic) markers that have been related to the diagnosis of T2DM according to ADA diagnosis criteria [21, 116] such as blood pressure, cholesterol levels, triglyceride levels, insulin resistance, serum insulin levels, fasting plasma glucose, 2-hour post glucose levels, HbA1c, several hormones (i.e., glucagon, growth hormone, etc.). We also included IGF-1 in diabetic markers [115].”

5.  Please note:  this comment in #4 also applies to descriptors for "T2DM markers," for example, "trunk and body fat mass" and "HBA1C" and "serum IGF1" are all included under T2DM markers; the rationale is unknown.

Reply

Similarly with the above comment, we have tried to present the variables of the “T2DM markers” in a consistent way, however, it was difficult based on the fact that the two columns (i.e., “Sarcopenic markers” and “T2DM markers”) should be synchronized (Table 2, “T2DM markers” column).

6.  Also, in Table 2, with a few exceptions, there are no comments on "Sarcopenic markers" that were measured but which did not change significantly.  This should also be noted. "

Reply

The reviewer is right, in the original manuscript there are only few comments about variables that were measured but they were not significant. In the revised manuscript (Table 2) we have added comments for the measured variable which however, did not change significant (Table 2, “Sarcopenic markers” column).

Round 2

Reviewer 2 Report

Overall, the authors revisions have improved the presentation of studies summarizing interventions on sarcopenia in T2DM.  The exception is in Table 1, in which the authors removed all studies which only assessed muscle mass or strength.  I strongly encourage the authors to NOT remove or exclude these important studies from the Table 1.

There is great value in listing these studies and data, in part because they support an important point (line 208):   "It is also evident that low valued muscle mass markers.... are strongly related to reduced glycemic control......"

My recommendation is to be consistent in the wording in the Table - for example, Kim et al., 2010 used SMI values, so the results statement: "SMI values were significantly decreased in patients with diabetes compared with control group" is appropriate, but "Patients with diabetes had three times higher risk of sarcopenia than control group" is not, because Kim et al only used muscle mass and not performance or strength in their definition of sarcopenia.   A more consistent presentation reflecting new science and consensus guidance criteria would be, e.g., "Patients with diabetes had three times higher risk of having a low SMI than control group..."  

I strongly encourage the authors to NOT remove or exclude these important studies from the Table 1, but be consistent in the use and definition of the term "Sarcopenia." 

Author Response

Reviewer 2

We would like to thank the reviewer for his/her effort and comments that led to the improvement of our manuscript.

Comment

Overall, the authors revisions have improved the presentation of studies summarizing interventions on sarcopenia in T2DM.  The exception is in Table 1, in which the authors removed all studies which only assessed muscle mass or strength.  I strongly encourage the authors to NOT remove or exclude these important studies from the Table 1.

There is great value in listing these studies and data, in part because they support an important point (line 208):   "It is also evident that low valued muscle mass markers.... are strongly related to reduced glycemic control......"

My recommendation is to be consistent in the wording in the Table - for example, Kim et al., 2010 used SMI values, so the results statement: "SMI values were significantly decreased in patients with diabetes compared with control group" is appropriate, but "Patients with diabetes had three times higher risk of sarcopenia than control group" is not, because Kim et al only used muscle mass and not performance or strength in their definition of sarcopenia.   A more consistent presentation reflecting new science and consensus guidance criteria would be, e.g., "Patients with diabetes had three times higher risk of having a low SMI than control group..."  

I strongly encourage the authors to NOT remove or exclude these important studies from the Table 1, but be consistent in the use and definition of the term "Sarcopenia."

Response

According to the reviewer’s suggestion we included in the Table 1 of the revised manuscript all the removed studies which only assessed muscle mass or strength (i.e., Kim et al. 2010, Kim et al. 2014; Leeder et al. 2013; Kalyani et al. 2015; Park et al. 2009). We would like to thank the reviewer for his/her suggestion because the revised Table 1 supports the relation between muscle mass markers and glycemic control which is pointed out in our review (Sarcopenia and T2DM, par. 5, lines 223-224).

Moreover, the wording in the Table was changed as recommended, and we believe that in the revised manuscript there is a consistency in presenting the studies included, while there is also a consistency in the use and definition of the term “Sarcopenia”.     

This manuscript is a resubmission of an earlier submission. The following is a list of the peer review reports and author responses from that submission.

Round 1

Reviewer 1 Report

Exercise and nutrition strategies for combating sarcopenia in 2 type 2 diabetes mellitus patients

It is a relevant topic to analyze both older adults’ prevalent health problems: DM and sarcopenia.

Please use older adults instead of elderly.

Abstract

– Need to be improved and there is lack of information.

Please include the aim of the study, some results and conclusion

Introduction

It is quite short and not enough to elucidate the gap in the scientific literature and the relevance of this study. Include at least more two paragraphs describing the "state of art" of this scientific question.

The aim of this study is not clear

Topic 2

There are lots of articles describing the definition of sarcopenia, so it is not necessary here since it is not the aim of this study. Table 1 is also not necessary.

Same for pathophysiology

Topic 3

Same commentary about topic 2. It is not necessary once that are lots of articles and books describing this.

Methods -

The authors should include Methodology in this article. The narrative review also needs to describe some methods about how the articles were found, which kind of study design were included, which electronic databases were consulted?

All this information has to be described.

The authors can see an example of the methods section of the literature review on the article https://onlinelibrary.wiley.com/doi/full/10.1111/obr.13088

Topic 4

The authors should deeply focus on describing the common pathophysiology of DM and sarcopenia and the connexion where one can compromise the other ones.

What is the objective of table 2? There are a miscellaneous of information and the aim of this table is not clear on their title or content.

This table should include effect size and precision via the width of confidence intervals regarding the risk of having sarcopenia in older adults individuals?

This table should describe more details of these studies:  the population, sample size, if only older adults, living in nurse care, community-dwelling or hospitalized, effect size and confidence interval, etc

Topic 5, table 3 and 4

This topic should focus only in include well-conducted clinical trials to show the effectiveness of exercise and diets to improve the parameters of sarcopenia and DM in older adults individuals who have both conditions. Unfortunately, there are different types of study designs and the individuals just have DM instead of DM and sarcopenia, then this table and results do not make sense to really bring a contribution.

There are lots of types of exercises and diets and then the authors should focus only in one type of intervention to bring relevant and complete information to contribute to the scientific literature.

The discussion is not enough to explore all the aspects.

The conclusion is not supported by the results due to the lack of clear have an aim and better-organized results and focus.

As the aim is not clear the article gets without a clear direction. The authors should keep the focus on one of the probable aims and go deeper into this. The great flaw of this article is the lack of methods that do not let clear any aspect of why all those articles are included and why lots of ones in the scientific literature are not included. However, the topic is relevant the article did not reach a good score in terms of scientific contribution and is very not well organized.

Author Response

Reviewer 1

We would like to thank the reviewer for his/her constructive criticism and valuable comments to improve the manuscript.

Exercise and nutrition strategies for combating sarcopenia and type 2 diabetes mellitus patients

It is a relevant topic to analyze both older adults’ prevalent health problems: DM and sarcopenia.

Please use older adults instead of elderly.

Response

According to the reviewer’s comment the “elderly” was changed to “older adults” throughout the manuscript.

Abstract

– Need to be improved and there is lack of information.

Please include the aim of the study, some results and conclusion

Response

The abstract section was improved by adding the aim of the study, results and conclusion.

Introduction

It is quite short and not enough to elucidate the gap in the scientific literature and the relevance of this study. Include at least more two paragraphs describing the "state of art" of this scientific question.

Response

The reviewer is right about the Introduction section. In the revised manuscript in the Introduction section were included information to support the rational of the present review. The revised Introduction section was almost re-written from scratch in order to highlight the “state of the art” of the present review. 

The aim of this study is not clear

 Response

The aim of the review is now highlighted in the last paragraph of the Introduction section and the relevant text now reads (Introduction section, par. 6, lines 111-117).

“To the best of our knowledge, there are no studies in the literature in which sarcopenia and T2DM were treated simultaneously. Therefore, the aim of this review is to evaluate techniques that has been applied for the treatment of sarcopenia and T2DM individually, which technics however could be used simultaneously in order to effectively deal with the two disorders. The present review is mainly focused on studies which has focused on exercise treatment techniques (aerobic, resistance, or combined aerobic and resistance) as well as diet interventions that is known to best combat the two disorders.”

Topic 2

There are lots of articles describing the definition of sarcopenia, so it is not necessary here since it is not the aim of this study. Table 1 is also not necessary.

Same for pathophysiology

Response

Based on the reviewer’s comments, the manuscript has been extensively revised, so the topics 2 and 3 were merged with the Introduction section highlighting the rational, the aim and the novelty of the review, while in the topic 2 is now described the methodology of the review (please see the comment below).

The Table 1 of the original review was omitted from the revised manuscript.

Topic 3

Same commentary about topic 2. It is not necessary once that are lots of articles and books describing this.

Response

As it was mentioned in the previous response to the reviewer, the manuscript has been extensively revised, while in the revised manuscript, the topics 2 and 3 were merged with the Introduction section. Moreover, in the topic 2 of the revised manuscript is described the methodology of the review (please see the comment below) (Introduction section).

Methods -

The authors should include Methodology in this article. The narrative review also needs to describe some methods about how the articles were found, which kind of study design were included, which electronic databases were consulted? All this information has to be described. The authors can see an example of the methods section of the literature review on the article https://onlinelibrary.wiley.com/doi/full/10.1111/obr.13088

Response

According to the correct reviewer’s suggestion, in the revised manuscript a new section was added the section “Methodology” where it is now described the steps that were followed for the studies selection for the present investigation. Now the relevant text reads (Methods section, page 3, lines 115-135):

“2. Methods

In the present review, were used studies using chronic exercise or/and diet inter-vention and clinical trials in which were included older adults suffering from sarcopenia or T2DM. We focused on treatment techniques that involved aerobic training, resistance training, or both trainings as well as diet interventions or nutrition analysis. Were also selected studies in which were used biomarkers of muscle mass and/or glycemic status.

An electronic search was performed using the following databases: PubMed and Google Scholar using the following search terms for each individual database:

1. “resistance training” OR “resistance exercise” OR “strength training” OR “aerobic exercise’’

2. “frailty” OR “muscle loss” OR “sarcopenia”

3. “muscle hypertrophy” OR “muscle strength” OR “skeletal muscle mass”

4. “diabetes melitus” OR “type 2 diabetes mellitus OR “insulin resistance”

5. “diet” OR “nutrition” OR “amino acids” OR “protein” OR “antioxidant” OR “life-style modification” OR “omega-3 supplementation” OR “vitamin D supplementa-tion”

6. “older adults” OR “elderly”

Recent literature published after 2000 was considered. We analyzed both titles and abstracts to decide whether a study could be included in our review. Articles were in-cluded if they fulfilled the following criteria: being peer-reviewed and/or research-based, met the search criteria and being considered well conducted studies.”  

Topic 4

The authors should deeply focus on describing the common pathophysiology of DM and sarcopenia and the connexion where one can compromise the other ones.

Response

According to the reviewer’s comment, in the revised manuscript a new section was added where the interaction between sarcopenia and T2DM is described. In the revised manuscript is also included the Table 1 where it is highlighted the close relation between sarcopenia and T2DM (Sarcopenia and T2DM section, page 3; Table 1, page 5).

What is the objective of table 2? There are a miscellaneous of information and the aim of this table is not clear on their title or content. This table should include effect size and precision via the width of confidence intervals regarding the risk of having sarcopenia in older adults individuals? This table should describe more details of these studies:  the population, sample size, if only older adults, living in nurse care, community-dwelling or hospitalized, effect size and confidence interval, etc

Response

Due to the removal of the original Table 1, the original Table 2 was changed in Table 1 in the revised manuscript. The aim of the Table 1 is to highlight the close relationship between sarcopenia and T2DM. More information about the population, the sample size and the participants’ living environment were added according to the reviewer’s suggestion (Table 1, page 5).

Topic 5, table 3 and 4

This topic should focus only in include well-conducted clinical trials to show the effectiveness of exercise and diets to improve the parameters of sarcopenia and DM in older adults individuals who have both conditions. Unfortunately, there are different types of study designs and the individuals just have DM instead of DM and sarcopenia, then this table and results do not make sense to really bring a contribution. There are lots of types of exercises and diets and then the authors should focus only in one type of intervention to bring relevant and complete information to contribute to the scientific literature.

Response

In the revised manuscript the original Tables 3 and 4 were merged into Table 2 and we have changed the structure of table according to the study design of each manuscript, the duration of the intervention, the type of exercise, the type of nutritional intervention, the sarcopenic and diabetes mellitus markers. Moreover, the content of the Table 2 has also been changed in order to include more relevant studies. Specifically, from the Table 2 were deleted the following studies in which the participants were not suffering from diabetes mellitus or sarcopenia while the participants were not older individuals or the diet intervention was generic:

Suetta, Charlotte, et al. "Resistance training induces qualitative changes in muscle morphology, muscle architecture, and muscle function in elderly postoperative patients." Journal of applied physiology 105.1 (2008): 180-186.

Candow, Darren G., et al. "Short-term heavy resistance training eliminates age-related deficits in muscle mass and strength in healthy older males." The Journal of Strength & Conditioning Research 25.2 (2011): 326-333.

Leenders, Marika, et al. "Elderly men and women benefit equally from prolonged resistance-type exercise training." Journals of Gerontology Series A: Biomedical Sciences and Medical Sciences 68.7 (2013): 769-779.

Greco, Marta, et al. "Early effects of a hypocaloric, Mediterranean diet on laboratory parameters in obese individuals." Mediators of Inflammation 2014 (2014).

Facchini, Francesco S., et al. "Relation between insulin resistance and plasma concentrations of lipid hydroperoxides, carotenoids, and tocopherols." The American journal of clinical nutrition 72.3 (2000): 776-779.

Valenzuela, Roxana E. Ruiz, et al. "Insufficient amounts and inadequate distribution of dietary protein intake in appar-ently healthy older adults in a developing country: implications for dietary strategies to prevent sarcopenia." Clinical interventions in aging 8 (2013): 1143.

Chan, Lin-Chien, et al. "Nutrition counseling is associated with less sarcopenia in diabetes: a cross-sectional and retro-spective cohort study." Nutrition (2021): 111269.

Takahashi, Fuyuko, et al. "Association between Geriatric Nutrition Risk Index and The Presence of Sarcopenia in People with Type 2 Diabetes Mellitus: A Cross-Sectional Study." Nutrients 13.11 (2021): 3729.

Furthermore, in the Table 2 of the revised manuscript we added new studies which are closer to the aim of the present review manuscript:

Kadoglou, N. P. E., et al. "The differential antiinflammatory effects of exercise modalities and their association with early carotid atherosclerosis progression in patients with type 2 diabetes." Diabetic Medicine 30.2 (2013): e41-e50.

Chen, HungTing, et al. "Effects of different types of exercise on body composition, muscle strength, and IGF1 in the el-derly with sarcopenic obesity." Journal of the American Geriatrics Society 65.4 (2017): 827-832.

Lustosa, Lygia P., et al. "Impact of resistance exercise program on functional capacity and muscular strength of knee ex-tensor in pre-frail community-dwelling older women: a randomized crossover trial." Brazilian Journal of Physical Therapy 15 (2011): 318-324

Seynnes, Olivier, et al. "Physiological and functional responses to low-moderate versus high-intensity progressive re-sistance training in frail elders." The Journals of Gerontology Series A: Biological Sciences and Medical Sciences 59.5 (2004): M503-M509.

Latham, Nancy K., et al. "A randomized, controlled trial of quadriceps resistance exercise and vitamin D in frail older people: the Frailty Interventions Trial in Elderly Subjects (FITNESS)." Journal of the American Geriatrics Society 51.3 (2003): 291-299.

Castaneda, Carmen, et al. "A randomized controlled trial of resistance exercise training to improve glycemic control in older adults with type 2 diabetes." Diabetes care 25.12 (2002): 2335-2341.

Tieland, Michael, et al. "Protein supplementation improves physical performance in frail elderly people: a randomized, double-blind, placebo-controlled trial." Journal of the American Medical Directors Association 13.8 (2012): 720-726.

Huang, JuiHua, et al. "Appropriate physical activity and dietary intake achieve optimal metabolic control in older type 2 diabetes patients." Journal of diabetes investigation 5.4 (2014): 418-427.

van Loon, Luc JC, et al. "Amino acid ingestion strongly enhances insulin secretion in patients with long-term type 2 dia-betes." Diabetes care 26.3 (2003): 625-630.

Moslehi, Nazanin, et al. "Associations between dairy products consumption and risk of type 2 diabetes: Tehran lipid and glucose study." International Journal of Food Sciences and Nutrition 66.6 (2015): 692-699.

Bischoff, Heike A., et al. "Muscle strength in the elderly: its relation to vitamin D metabolites." Archives of physical med-icine and rehabilitation 80.1 (1999): 54-58.

Semba, Richard D., et al. "Carotenoid and vitamin E status are associated with indicators of sarcopenia among older women living in the community." Aging clinical and experimental research 15.6 (2003): 482-487.

The discussion is not enough to explore all the aspects.

Response

In the revised manuscript the “Discussion” section was changed to “Discussion and conclusions” section and it was greatly changed in order to explore all the aspects of the review (Discussion and conclusion section, page 13, lines 380-525).

The conclusion is not supported by the results due to the lack of clear have an aim and better-organized results and focus.

Response

According to the reviewer’s comments we have reorganized the entirely manuscript in order to support the aim and the conclusion of the manuscript in order to clearly present the results derived from the papers that were used. Regarding conclusion section, in the revised manuscript were merged with the Discussion section. Now the relevant text reads (Discussion and conclusion section, par. 9, page 15, lines 509-525):

“To the best of our knowledge, studies that combine both exercise and nutritional interventions are mainly focused on obese people [97 - 100]. Apart from that, there exists very limited research on proposing methods to simultaneously compensate for both of those common diseases. In a relevant study, diabetic mice were supplemented with ginseng diet and followed aerobic exercise, were found to have improvements in muscle mass and on insulin resistance [101]. Generally, while there is an agreement that both diseases should be treated simultaneously, researchers seem to ignore people suffering from both diseases and prefer to examine the treatment techniques separately. Never-theless, sarcopenia and diabetes mellitus have high prevalence among the older adults which in turn may lead to low quality of life. The available data show that a combination of supervised combined training along with protein intake > 1gr/kg/d seems to be the safest treatment, if not the best, in order to compensate both sarcopenia and T2DM. Of note, nutrition counseling is also important to achieve diet adherence [28, 102]. Due to the fact that the one disorder fuels the other, more studies are needed involving people suffering both disorders. Since sarcopenia and T2DM have been widespread among population in modern era, it is imperative to find the best combination of exercise and diet interventions that will be succeed in managing sarcopenia and diabetes mellitus.”

As the aim is not clear the article gets without a clear direction. The authors should keep the focus on one of the probable aims and go deeper into this. The great flaw of this article is the lack of methods that do not let clear any aspect of why all those articles are included and why lots of ones in the scientific literature are not included. However, the topic is relevant the article did not reach a good score in terms of scientific contribution and is very not well organized.

Response

Based on the reviewer’s comments, the manuscript has been extensively revised and we have clearly highlighted the aim of the review. Moreover, we have included a “Methodology” section with information about the criteria for the studies that were included. Additionally, in the revised manuscript we have changed the overall organization according to the reviewer’s comments and we hope that in its present form the scientific contribution of the review will be sound. 

Reviewer 2 Report

Dear Authors:

Regarding the manuscript with title Exercise and nutrition strategies for combating sarcopenia in type 2 diabetes mellitus patients” I have one major concern. Also minor comments were addressed.

Major comment:

There are plenty of systematic reviews regarding the interaction between “sarcopenia and exercise”; “type 2 diabetes and exercise”; “diet and sarcopenia” and “diet and type 2 diabetes”. On this review, authors present on Table 2 and 3 sources of evidence with lower quality of evidence.

Minor comments:

Comment 1:

Regarding what is stated on this review, I suggest authors to change the title from “Exercise and nutrition strategies for combating sarcopenia in  type 2 diabetes mellitus patients” to “Exercise and nutrition strategies for combating sarcopenia and type 2 diabetes mellitus”

Comment 2:

On keywords, authors must delete the expression “muscle loss”, as i tis represented by the word “sarcopenia”

Comment 3:

Lines 14-16: Authors must change “exercise and nutrition as countermeasures for combating sarcopenia and T2DM either separately or based on their bidirectional relationship. Lifestyle changes such as exercise and diet” by “exercise and balanced diet as countermeasures for combating sarcopenia and T2DM either separately or based on their bidirectional relationship. Lifestyle changes such as exercise and balanced diet”

Comment 4:

On Abstract, authors must clarify the main purposes of this review

Comment 5:

Lines 16-18: Authors must delete the sentence “However, an active debase exists in the literature regarding the effects of current protein intake recommendations in elderly suffering chronic diseases” as this sentence refers to a very specific quastion regarding diet.

Comment 6:

Lines 29-30: Authors have to add references to the following sentence: “Recent studies suggested that sarcopenia may not only be a cause for diabetes but also a consequence, indicating a bidirectional relationship between the two conditions”

Comment 7:

Line 33-34: Authors must change “Nutrition and lifestyle” by  “Lifestyle changes”.

Comment 8:

Lines 35-36: Authors must delete the sentence “Especially, protein supplementation has been suggested to help not only in body weight maintenance but also in regulation of glucose homeostasis [5, 6]” for the same reason stated on Comment 4.

Comment 9:

Line 38: Authors must change “exercise and nutritional strategies as countermeasures for sarcopenia” by “exercise and nutritional strategies as countermeasures for sarcopenia and T2D”

Comment 10:

Lines 135-136: I do not understand the following sentence: “Elderly suffering from T2DM may tolerate reduced muscle mass, lower muscle strength and body function compared to their healthy counterparts.”

Comment 11:

Line 299: Authors must change “both of disorders” by “both disorders”

Comment 12:

Line 301-302: Authors must change “in both conditions while, dietary interventions with supplements of protein, vitamin D, and omega-3 fatty acids” by “in both conditions, while on dietary interventions, supplements of protein, vitamin D, and omega-3 fatty acids”

Comment 13:

Line 311: Authors must change “separately the treatment techniques” by “the treatment techniques separately”

Comment 14:

Lines 312-313: Therefore, it is imperative to find the best combination of exercise and diet interventions that will prevent sarcopenia and diabetes mellitus. Authors are focused on the prevention or management of these two diseases?

Author Response

Reviewer 2

Regarding the manuscript with title Exercise and nutrition strategies for combating sarcopenia in type 2 diabetes mellitus patients” I have one major concern. Also minor comments were addressed.

We would like to thank the reviewer for his/her constructive criticism and valuable comments to improve the manuscript.

Major comment:

There are plenty of systematic reviews regarding the interaction between “sarcopenia and exercise”; “type 2 diabetes and exercise”; “diet and sarcopenia” and “diet and type 2 diabetes”. On this review, authors present on Table 2 and 3 sources of evidence with lower quality of evidence.

Response

In the revised manuscript the original Tables 3 and 4 were merged into Table 2 and we have changed the structure of table according to study design, duration of the intervention, type of exercise, type of nutritional intervention, sarcopenic and diabetes mellitus markers. Moreover, the content of the Table 2 has also changed in order to include more relevant studies. Specifically, from the Table 2 were deleted the following studies in which the participants were not suffering from diabetes mellitus or sarcopenia while they were not older individuals or the diet intervention was generic:

Suetta, Charlotte, et al. "Resistance training induces qualitative changes in muscle morphology, muscle architecture, and muscle function in elderly postoperative patients." Journal of applied physiology 105.1 (2008): 180-186.

Candow, Darren G., et al. "Short-term heavy resistance training eliminates age-related deficits in muscle mass and strength in healthy older males." The Journal of Strength & Conditioning Research 25.2 (2011): 326-333.

Leenders, Marika, et al. "Elderly men and women benefit equally from prolonged resistance-type exercise training." Journals of Gerontology Series A: Biomedical Sciences and Medical Sciences 68.7 (2013): 769-779.

Greco, Marta, et al. "Early effects of a hypocaloric, Mediterranean diet on laboratory parameters in obese individuals." Mediators of Inflammation 2014 (2014).

Facchini, Francesco S., et al. "Relation between insulin resistance and plasma concentrations of lipid hydroperoxides, carotenoids, and tocopherols." The American journal of clinical nutrition 72.3 (2000): 776-779.

Valenzuela, Roxana E. Ruiz, et al. "Insufficient amounts and inadequate distribution of dietary protein intake in appar-ently healthy older adults in a developing country: implications for dietary strategies to prevent sarcopenia." Clinical interventions in aging 8 (2013): 1143.

Chan, Lin-Chien, et al. "Nutrition counseling is associated with less sarcopenia in diabetes: a cross-sectional and retro-spective cohort study." Nutrition (2021): 111269.

Takahashi, Fuyuko, et al. "Association between Geriatric Nutrition Risk Index and The Presence of Sarcopenia in People with Type 2 Diabetes Mellitus: A Cross-Sectional Study." Nutrients 13.11 (2021): 3729.

Furthermore, in the Table 2 of the revised manuscript we added new studies which are closer to the aim of the present review manuscript:

Kadoglou, N. P. E., et al. "The differential antiinflammatory effects of exercise modalities and their association with early carotid atherosclerosis progression in patients with type 2 diabetes." Diabetic Medicine 30.2 (2013): e41-e50.

Chen, HungTing, et al. "Effects of different types of exercise on body composition, muscle strength, and IGF1 in the el-derly with sarcopenic obesity." Journal of the American Geriatrics Society 65.4 (2017): 827-832.

Lustosa, Lygia P., et al. "Impact of resistance exercise program on functional capacity and muscular strength of knee ex-tensor in pre-frail community-dwelling older women: a randomized crossover trial." Brazilian Journal of Physical Therapy 15 (2011): 318-324

Seynnes, Olivier, et al. "Physiological and functional responses to low-moderate versus high-intensity progressive re-sistance training in frail elders." The Journals of Gerontology Series A: Biological Sciences and Medical Sciences 59.5 (2004): M503-M509.

Latham, Nancy K., et al. "A randomized, controlled trial of quadriceps resistance exercise and vitamin D in frail older people: the Frailty Interventions Trial in Elderly Subjects (FITNESS)." Journal of the American Geriatrics Society 51.3 (2003): 291-299.

Castaneda, Carmen, et al. "A randomized controlled trial of resistance exercise training to improve glycemic control in older adults with type 2 diabetes." Diabetes care 25.12 (2002): 2335-2341.

Tieland, Michael, et al. "Protein supplementation improves physical performance in frail elderly people: a randomized, double-blind, placebo-controlled trial." Journal of the American Medical Directors Association 13.8 (2012): 720-726.

Huang, JuiHua, et al. "Appropriate physical activity and dietary intake achieve optimal metabolic control in older type 2 diabetes patients." Journal of diabetes investigation 5.4 (2014): 418-427.

van Loon, Luc JC, et al. "Amino acid ingestion strongly enhances insulin secretion in patients with long-term type 2 dia-betes." Diabetes care 26.3 (2003): 625-630.

Moslehi, Nazanin, et al. "Associations between dairy products consumption and risk of type 2 diabetes: Tehran lipid and glucose study." International Journal of Food Sciences and Nutrition 66.6 (2015): 692-699.

Bischoff, Heike A., et al. "Muscle strength in the elderly: its relation to vitamin D metabolites." Archives of physical med-icine and rehabilitation 80.1 (1999): 54-58.

Semba, Richard D., et al. "Carotenoid and vitamin E status are associated with indicators of sarcopenia among older women living in the community." Aging clinical and experimental research 15.6 (2003): 482-487.

Minor comments:

Comment 1:

Regarding what is stated on this review, I suggest authors to change the title from “Exercise and nutrition strategies for combating sarcopenia in  type 2 diabetes mellitus patients” to “Exercise and nutrition strategies for combating sarcopenia and type 2 diabetes mellitus”

Response

The title was changed according to the reviewer’s suggestion. However, we added the phrase “older adults” in order to describe the age group of the participants. The title of the revised manuscript is now reads:

“Exercise and nutrition strategies for combating sarcopenia and type 2 diabetes mellitus in older adults”

Comment 2:

On keywords, authors must delete the expression “muscle loss”, as it is represented by the word “sarcopenia”

Response

The words “muscle loss” were removed from the key words as suggested. 

Comment 3:

Lines 14-16: Authors must change “exercise and nutrition as countermeasures for combating sarcopenia and T2DM either separately or based on their bidirectional relationship. Lifestyle changes such as exercise and diet” by “exercise and balanced diet as countermeasures for combating sarcopenia and T2DM either separately or based on their bidirectional relationship. Lifestyle changes such as exercise and balanced diet”

Response

The sentence was changed as suggested by the reviewer. The relevant text in the revised manuscript is now reads (Abstract, lines 16-18):

“Exercise and balanced diet are used as effective countermeasures for combating sarcopenia and T2DM in older adults based on their bidirectional relationship. Lifestyle changes such as exercise and balanced diet seem to play an important role in the remission of the diseases.”

Comment 4:

On Abstract, authors must clarify the main purposes of this review

Response

The aim of the review is clarified in the revised manuscript (Abstract, lines 13-16):

“The aim of the present review is to present techniques that has been applied for the treatment of sarcopenia and T2DM separately, which in turn could be used simultaneously in order to deal with the two disorders.”

Comment 5:

Lines 16-18: Authors must delete the sentence “However, an active debase exists in the literature regarding the effects of current protein intake recommendations in elderly suffering chronic diseases” as this sentence refers to a very specific quastion regarding diet.

Response

The sentence was removed from the revised manuscript as the reviewer suggested.

Comment 6:

Lines 29-30: Authors have to add references to the following sentence: “Recent studies suggested that sarcopenia may not only be a cause for diabetes but also a consequence, indicating a bidirectional relationship between the two conditions”

Response

Two references were added to this statement (Introduction, par. 4, lines 88-90):

23.       Kim, Tae Nyun, et al. "Prevalence and determinant factors of sarcopenia in patients with type 2 diabetes: the Korean Sarcopenic Obesity Study (KSOS)." Diabetes care 33.7 (2010): 1497-1499.

24.       Souza, Anelza Biene Farias, et al. "Association between sarcopenia and diabetes in community dwelling elderly in the Amazon region–Viver Mais Project." Archives of gerontology and geriatrics 83 (2019): 121-125.

Comment 7:

Line 33-34: Authors must change “Nutrition and lifestyle” by  “Lifestyle changes”.

Response

The phase “Nutrition and lifestyle” was changed to “Lifestyle changes” in the revised manuscript (Introduction, par. 4, lines 93-95).

Comment 8:

Lines 35-36: Authors must delete the sentence “Especially, protein supplementation has been suggested to help not only in body weight maintenance but also in regulation of glucose homeostasis [5, 6]” for the same reason stated on Comment 4.

Response

The sentence was deleted from the revised manuscript.

Comment 9:

Line 38: Authors must change “exercise and nutritional strategies as countermeasures for sarcopenia” by “exercise and nutritional strategies as countermeasures for sarcopenia and T2D”

Response

Due to extensive revision of the manuscript, the particular point was not included in the revised paper. However, the proposed change was adopted in another part of the review (i.e., Abstract, lines 16-17).  

Comment 10:

Lines 135-136: I do not understand the following sentence: “Elderly suffering from T2DM may tolerate reduced muscle mass, lower muscle strength and body function compared to their healthy counterparts.”

Response

Due to extensive revision of the manuscript, this sentence was not included in the revised manuscript. 

Comment 11:

Line 299: Authors must change “both of disorders” by “both disorders”

Response

The phrase “both of disorders” was changed to “both disorders” as suggested throughout the manuscript (lines 241, 385, 389, 511, 514, 515, 522).

Comment 12:

Line 301-302: Authors must change “in both conditions while, dietary interventions with supplements of protein, vitamin D, and omega-3 fatty acids” by “in both conditions, while on dietary interventions, supplements of protein, vitamin D, and omega-3 fatty acids”

Response

Due to extensive revision of the manuscript, this sentence was not included in the revised manuscript. 

Comment 13:

Line 311: Authors must change “separately the treatment techniques” by “the treatment techniques separately”

Response

The phrase “separately the treatment techniques” was changed to “to examine the treatment techniques separately” as suggested (Discussion and conclusion section, par. 9, line 516).

Comment 14:

Lines 312-313: Therefore, it is imperative to find the best combination of exercise and diet interventions that will prevent sarcopenia and diabetes mellitus. Authors are focused on the prevention or management of these two diseases?

Response

In the revised manuscript present in a clearer way the aim of the present review manuscript (Abstract, lines 13-16; Introduction, par. 6, lines 111-117).

Moreover, in the revised manuscript we have reorganize the entire text in order to highlight that we focus on the management of these two diseases (i.e., sarcopenia and T2DM). 

Round 2

Reviewer 1 Report

Dear authors,

You are presenting a new article after all my considerations in the first review. As it is a completely new article with a new title, aim, text, discussion, and tables. It seems that the article improves after the first version, however, it is a  completely new article. Considering this I strongly recommend the authors start a new process of submission. This new version is a new article and then, it is necessary that the editor and pairs revision read I include a new evaluation process.

Best regards,

Reviewer 2 Report

Dear Authors:

Regarding the manuscript with title Exercise and nutrition strategies for combating sarcopenia in type 2 diabetes mellitus patients” I still have the same major concern previously stated. Also minor comments were addressed.

Major Comments

Regarding Table 1, authors must only use high quality studies, such as systematic reviews:

“Type 2 Diabetes Mellitus is Associated with Increased Risk of Sarcopenia: A Systematic Review and Meta-analysis”

“The prevalence and risk factors of sarcopenia in patients with type 2 diabetes mellitus: a systematic review and meta-analysis”.

For sure in these systematic reviews, the studies that authors presented in the manuscript are included.

Regarding Table 2, as I mentioned in my previous review, there are plenty of systematic reviews regarding the interaction between “sarcopenia and exercise”; “type 2 diabetes and exercise”; “diet and sarcopenia” and “diet and type 2 diabetes”. These systematic reviews must be used on Table 2 and not lowest level of evidence studies.

Minor Comments:

Comment 1:

Lines 13-16: Authors must change “The aim  of the present review is to present techniques that has been applied for the treatment of sarcopenia and T2DM separately, which in turn could be used simultaneously in order to deal with the two disorders” by “The aims  of the present review are: 1) present diet and exercise interventions for the management of sarcopenia and T2DM and 2) which diet and exercise interventions can be used simultaneously in order to effectively deal with these two disorders

Comment 2:

On line 23, Authors must change “to manage the sarcopenia and T2DM and sarcopenia” by “to manage the sarcopenia and T2DM”

Comment 3:

Therefore, the aim of this review is to evaluate techniques that has been applied for the treatment of sarcopenia and T2DM individually, which technics however could be used simultaneously in order to effectively deal with the two disorders. The present review is mainly focused on studies which has focused on exercise treatment techniques (aerobic, resistance, or combined aerobic and resistance) as well as diet interventions that is known to best combat the two disorders” by “Therefore, the aim of this review is to evaluate diet and exercise interventions that has been applied for the treatment of sarcopenia and T2DM individually and which exercise and diet interventions can be used simultaneously in order to effectively deal with the two disorders”.